# Multi-Objective Artificial Bee Colony Algorithm Based on Scale-Free Network for Epistasis Detection

**DOI:** 10.3390/genes13050871

**Published:** 2022-05-12

**Authors:** Yijun Gu, Yan Sun, Junliang Shang, Feng Li, Boxin Guan, Jin-Xing Liu

**Affiliations:** School of Computer Science, Qufu Normal University, Rizhao 276826, China; guyijun216@163.com (Y.G.); sunyan225@126.com (Y.S.); lifeng_10_28@163.com (F.L.); neuguanboxin@163.com (B.G.); sdcavell@126.com (J.-X.L.)

**Keywords:** artificial bee colony, scale-free network, epistasis detection, single nucleotide polymorphism, complex disease

## Abstract

In genome-wide association studies, epistasis detection is of great significance for the occurrence and diagnosis of complex human diseases, but it also faces challenges such as high dimensionality and a small data sample size. In order to cope with these challenges, several swarm intelligence methods have been introduced to identify epistasis in recent years. However, the existing methods still have some limitations, such as high-consumption and premature convergence. In this study, we proposed a multi-objective artificial bee colony (ABC) algorithm based on the scale-free network (SFMOABC). The SFMOABC incorporates the scale-free network into the ABC algorithm to guide the update and selection of solutions. In addition, the SFMOABC uses mutual information and the K2-Score of the Bayesian network as objective functions, and the opposition-based learning strategy is used to improve the search ability. Experiments were performed on both simulation datasets and a real dataset of age-related macular degeneration (AMD). The results of the simulation experiments showed that the SFMOABC has better detection power and efficiency than seven other epistasis detection methods. In the real AMD data experiment, most of the single nucleotide polymorphism combinations detected by the SFMOABC have been shown to be associated with AMD disease. Therefore, SFMOABC is a promising method for epistasis detection.

## 1. Introduction

Genome-wide association studies (GWAS) play a significant role in determining the genetic mechanisms of complex diseases [1,2]. With the advent of high-throughput sequencing technology, large numbers of single nucleotide polymorphisms (SNPs) have been identified. Through tremendous advances in gene localization, SNPs have been accepted as commonly used markers of human genetic variation, and SNP interactions, or epistasis, are important factors that affect disease incidence. In other words, the genetic mechanisms of complex diseases can be better understood through these SNP data and SNP interactions [3,4,5,6]. However, epistasis detection faces challenges such as high dimensionality and a small data sample size [7,8,9].

To address these challenges, a series of SNP interaction detection methods have been proposed. For example, Wan et al. [10] proposed the learning method SNPRuler based on predictive rule inference to discover epistatic interactions associated with disease, and they also proposed a classical method, BOOST, based on categorization to identify SNP interactions [11]. However, these methods have several shortcomings, such as high time complexity, low optimization efficiency, and fast convergence. To solve these problems, a large number of epistasis detection methods have been proposed in the past ten years. For example, Christian et al. [12] analyzed the runtime and detection power of these methods, and Shang et al. [13] provided a comprehensive review of the methods based on the ant colony optimization (ACO) algorithm. Wang et al. [14] proposed a two-stage ant colony optimization (ACO) algorithm, named AntEpiSeeker, for epistasis detection. In AntEpiSeeker, ACO is used to obtain a predefined number of highly suspected SNP sets in the first stage, and the final solution is obtained by an exhaustive search in the second stage. Sun et al. [15] proposed the EACO method, based on the ACO algorithm, which introduces heuristic information and uses two objective functions to detect epistatic interactions. Zhang et al. [16] proposed a selective information particle swarm optimization algorithm (SIPSO), which introduces the scale-free networks as its population structure and uses the mutual information (MI) as the objective function to evaluate SNP interactions. Aflakparast et al. [17] proposed a Cuckoo search epistasis (CSE) method, which uses Bayesian scoring as the objective function and combines this with the CSE algorithm to detect epistasis.

The swarm intelligence algorithm has gradually become an effective means to solve epistasis problems in recent years. Tuo [18] proposed a fast method, FDHE-IW, for detecting high-order epistatic interactions based on an interaction weight. FDHE-IW uses the symmetric uncertainty (SU) value as the objective function, selects the top k SNP combinations based on the SU value of each SNP, and then uses forward searching to select higher-order SNP combinations. Tuo et al. [19] proposed a multi-population harmony search (HS) algorithm dedicated to the detection of high-order SNP interactions (MP-HS-DHSI). It consists of three stages. In the first stage, a multi-objective HS algorithm is used to discover the candidate SNP combinations. In the second and third stages, the G-test statistical method and multifactor dimensionality reduction (MDR) are used to verify the authenticity of the candidate solutions. Chen et al. [20] proposed a multi-objective genetic algorithm (EpiMOGA) for SNP interaction detection that uses the K2-Score and the Gini index as the objective functions. Pashaei et al. [21] proposed a new hybrid approach that combines the strengths of two existing metaheuristics: the binary dragonfly algorithm and the binary black hole algorithm (BBHA). The swarm intelligence algorithm has been shown to perform well in epistasis detection. The artificial bee colony algorithm (ABC) is a new swarm intelligence algorithm that was inspired by the swarm foraging behavior of bees [22].

ABC has been introduced in recent years for epistasis detection in GWAS. Chen et al. [23] proposed an epistasis mining approach based on an ABC optimized Bayesian network (BnBeeEpi). BnBeeEpi used two Bayesian network (BN) scoring functions and introduced the decomposable *BIC* score to solve the problem of large-scale network learning. Guan et al. [24] proposed a random grouping-based self-regulating artificial bee colony algorithm for epistasis detection. RCABC uses a dynamic random grouping (DRG) strategy to decompose all features in a dataset and then uses a self-regulating bee colony optimizer to detect relevant interactive features in each subset. Li et al. [25] proposed an epistatic interaction multi-objective ABC algorithm based on decomposition (EIMOABC/D), which uses a rank probability model and a local search strategy to address the problems in GWAS.

Compared with other swarm intelligence algorithms, the ABC algorithm has fewer control parameters and a simpler structure [26]. However, the existing ABC-based epistasis detection methods face the following challenges: ABC suffers from a slow convergence problem, such methods easily fall into the local optimum during the iterative process, and the single-objective strategy cannot effectively evaluate the epistatic model. Therefore, the research motivations of this study were as follows: to improve the convergence problem of the ABC, to introduce a random strategy to avoid ABC falling into the local optimum, and to select the appropriate objective functions to effectively evaluate the epistasis model.

Based on the above discussion and findings, this study proposed a multi-objective ABC algorithm based on the scale-free network (SFMOABC) for epistasis detection. We carried out experiments on 12 small-scale and 12 large-scale simulation models and a real age-related macular disease (AMD) dataset. The results show that SFMOABC is more effective for epistasis detection than the other methods used for the comparison. The contributions of the SFMOABC are summarized as follows: (1) a mechanism that adopts the scale-free network to guide the search of the ABC. The scale-free network has the characteristics of power law distribution and a low degree-degree correlation coefficient. The characteristics of the scale-free network can help each employed bee to learn more effective information from its neighbors, which improves the detection power; (2) the multi-objective strategy in which the MI and K2-score are used to characterize various epistasis models and improve the detection power; and (3) the opposition-based learning strategy. ABC can easily to fall into the local optimum as the iteration progresses, and the opposition-based learning strategy improves the detection power by improving the randomness of the algorithm.

## 2. Methods

### 2.1. Scale-Free Network

The concept of scale-free networks was introduced in a paper published by Albert-Laszlo Barabasi and Reka Albert [27]. The scale-free network is a complex network model where node degree distribution is approximately a power law distribution. The power law distribution of the node degree is intuitively shown as follows: most “ordinary” nodes have few connections, whereas a few “hot” nodes have many connections. Such a network is called a scale-free network, and “hot” nodes in the network are called hub nodes. This phenomenon can be described as the following formula [27]:(1)$$P\left(k\right)∼k^{-\gamma }$$
where $p\left(k\right)$ is the probability that any node owns degree k in the network. γ is a parameter describing the network structure, and its value range is usually 2 to 3.

The scale-free network is often compared with the random network, which is a network made up of nodes that are randomly connected. The degree of each node in the random network is similar, and there is no hub node. The Comparison diagram of the scale-free network and the random network is shown in Figure 1.

Additionally, Barabasi and Reka Albert proposed the classical BA model for constructing the scale-free network. The specific structure of the BA model is as follows:(1)Growth: Start with a small fully connected network $G_{0}$
which has $m_{0}$ nodes, and gradually add new nodes one at a time.(2)Connection: Assume that the original network already has m
nodes $\left(s_{1},s_{2},\cdots ,s_{m}\right)$. When a new node $s_{m+1}$ is added, it connects n links to the original m nodes, where $n<m_{0}$.(3)Priority connection: The connection strategy gives priority to the nodes with a higher degree. For an original node $s_{i}$
$\left(1\leq i\leq m\right)$, the probability $P_{i}$ that the new node is connected to it can be described as
(2)$$P_{i}=\frac{d_{i}}{\sum_{j=1}^{m}d_{j}}$$
where $d_{i}$ is the degree of node $s_{i}$ in the original network, and $d_{j}$ is the degree of node $s_{j}$ in the original network

### 2.2. Artificial Bee Colony Algorithm

The ABC algorithm is a new global optimization algorithm based on swarm intelligence, which is usually used to solve numerical optimization problems [26,28,29]. It is inspired by the honey gathering behavior of bees. To find the optimal solution to a problem, bees carry out different activities according to their respective divisions of labor and share information with each other. The ABC algorithm consists of three bee types: employed bees, onlooker bees, and scout bees. Among them, the number of employed bees is equal to the number of onlooker bees. The employed bees are responsible for exploring new food sources and sharing information about food sources with the onlooker bees. According to the shared information, onlooker bees make choices about the food sources. Scout bees discard the food sources according to certain rules and then look for new ones.

Suppose the solution to the optimization problem has D dimensions, the number of food sources is N, and the number of employed bees is consistent with the number of food sources. The standard ABC algorithm regards the process of solving optimization problems as searching the D-dimensional solutions in the search space. Each food source represents a possible solution to the problem, and the amount of nectar in the food source corresponds to the fitness value of the corresponding solution. The food source is expressed as $x_{i}=(x_{i1},x_{i2},\cdots ,x_{iD})$, where i=1,2,⋯,N. During the initialization stage, the food source $x_{i}$ can be generated according to the following formula:(3)$$x_{ij}=x_{j}^{\min}+rand\left(0,1\right)\left(x_{j}^{\max}-x_{j}^{\min}\right)$$
where i={1,2,⋯,N}, j={1,2,⋯,D}, $x_{j}^{\max}$ and $x_{j}^{\min}$ are the upper and lower boundaries of j dimension, and $rand\left(0,1\right)$ represents a random number uniformly distributed between 0 and 1.

After the initialization stage, the employed bees search for the new food sources by changing their current positions, which can be described as
(4)$$v_{ij}=x_{ij}+rand\left(-1,1\right)\left(x_{ij}-x_{kj}\right)$$
where k is a random value that satisfies the condition $k\in \{1,2,\cdots ,SN\}\;\left(k\neq i\right)$, and $rand\left(-1,1\right)$ represents a random number uniformly distributed between −1 and 1. The new food source $v_{i}$ is evaluated, and a greedy strategy is conducted to compare the new food source and the original food source. If the new food source is better than the old one, the employed bees will remember the location of the new food source. Otherwise, the employed bees will keep the original food source.

When all employed bees have completed the search process, the onlooker bees collect information from the employed bees and select food sources according to the probability value $P_{i}$ associated with food source $v_{i}$. The probability value can be calculated using the following equation:(5)$$P_{i}=\frac{fit_{i}}{\sum_{i=1}^{N}fit_{i}}$$
where $fit_{i}$ is the fitness value of food source $v_{i}$. The onlooker bees use a roulette strategy to select the food sources found by the employed bees. This means that the higher the fitness value of the food source is, the more likely it is to be selected. Taking the optimization problem of minimization function as an example, the fitness function of the food source is defined as
(6)$$fit_{i}=\left\{\begin{array}{l} \frac{1}{1+f_{i}}\;f_{i}\geq 0\\ 1+abs\left(f_{i}\right)\;otherwise \end{array}\right.$$
where $f_{i}$ is the cost value of solution $v_{i}$. If a solution is not selected, the onlooker bee will discard it and generate a new solution through (4).

In the scout stage, the ABC checks the parameter *limit* to decide whether to discard the food source. When the employed bee fails to find a better food source after *limit* iterations, it discards this food source. Then, the employed bee turns into the scout bee and randomly generates a new solution to replace the original solution according to Formula (3).

### 2.3. Multi-Objective Artificial Bee Colony Algorithm Based on the Scale-Free Network

#### 2.3.1. Objective Function

To improve the detection power of the algorithm, two objective functions, mutual information (MI) and Bayesian network (BN) scoring, are used in the SFMOABC.

The first objective function is MI, which is a measure based on information entropy that is used to evaluate the uncertainty between variables [30,31]: the higher the MI value, the stronger the correlation between the SNP combination and the phenotype.

The MI between the SNP combination and the phenotype can be described as
(7)$$MI\left(S;Y\right)=H\left(S\right)+H\left(Y\right)-H\left(S;Y\right)$$
where S is the SNP combination, Y is the phenotype, $H\left(S\right)$ is the entropy of S, $H\left(Y\right)$ is the entropy of Y, and H(S;Y) represents the joint entropy of S and Y.

The second objective function is the K2-Score based on the BN, which is used to evaluate the dependence of variables [32,33]. The lower the value of the K2-Score, the greater the correlation between the SNP combination and the phenotype. The BN model is a probabilistic graph model that can be expressed by a directed acyclic graph $G=\left(V,\;E\right)$. In the directed acyclic graph, the node set V is composed of random variables, and E is a set of edges. The BN model represents causality by connecting the edges between nodes and the conditional dependence between two connected variables. The K2-Score based on the BN is described as
(8)$$K2=\prod_{i=1}^{I}\left[\frac{\left(J-1\right)!}{\left(n_{i}+J-1\right)!}\prod_{j=1}^{J}n_{ij}!\right]$$
where I is the number of combinations of SNP nodes with different values, J is the number of states of phenotypic nodes, $n_{i}$ is the number of cases for the i-th combination, and $n_{ij}$ represents the number of cases for the j-th phenotype at the i-th disease node.

To simplify the calculation, Formula (9) is usually converted into logarithmic form, which can be rewritten as
(9)$$K2_{\log}=\sum_{i=1}^{I}\left[\sum_{k}^{n_{}+1}\log\left(k\right)-\sum_{j=1}^{J}\sum_{s=1}^{n_{}}\log\left(s\right)\right]$$

In the SFMOABC, the two objective functions are integrated to evaluate the correlation between the SNP combination and the phenotype using multiple aspects in a novel format, which can be defined as [34]
(10)$$fit=\frac{MI}{K2}$$

It can be seen that the larger the fit value is, the stronger the correlation between the SNP combination and the phenotype will be.

#### 2.3.2. Initialization Based on the Scale-Free Network

In the initialization stage, the SFMOABC generates an initial population with N solutions (food sources), and each food source represents an SNP combination. Then, the SFMOABC calculates the fitness value of each food source in the population through Formula (10) and sorts these candidate solutions in descending order according to their fitness values. Meanwhile, the BA algorithm is used to construct a scale-free network, and the total number of nodes in the network is consistent with the number of food sources in the initial population.

During network construction, each node needs to be numbered. For example, the total number of nodes in the network is N, and the number of hub nodes is $m_{0}$. First, the hub nodes are numbered in a range from 1 to $m_{0}$, and the order is random. Then, the remaining $N-m_{0}$ nodes are numbered sequentially from $m_{0}+1$ according to the order that they joined the network. After all nodes have been numbered, a complete scale-free network is constructed. Each solution in the population corresponds to a node in the scale-free network. The first $m_{0}$ solutions in descending order according to their fitness values correspond to the hub nodes of the network, and these solutions are called elite solutions (elite food sources). The remaining solutions correspond to the nodes numbered from $m_{0}+1$ to N, which are called normal solutions (normal food sources). 

#### 2.3.3. Solution Updating Based on the Scale-Free Network

The SFMOABC relies on both the scale-free network and the opposition-based learning strategy to update solutions representing SNP combinations. Both the employed bee and onlooker bee stages of the SFMOABC involve solution updating. The scale-free network is a network model with a power law distribution and a low degree-degree correlation coefficient. Based on these characteristics, low-quality solutions are more likely to move closer to high-quality solutions. Therefore, when the solution is updated, the SFMOABC will find the solution corresponding to its neighbor with the largest degree in the network and then move closer to it. However, the quality of the hub nodes is high. When any two hub nodes are close to each other during the updating process, the SFMOABC can easily fall into the local optimum. Therefore, in order to increase the exploration ability and prevent premature convergence, a solution is randomly selected from the solution space when the hub code is updated.

In the initialization stage, the solutions are sorted in descending order according to their fitness values. They are then divided into two categories: elite solutions and normal solutions. These two types of solution participate in each iteration of the SFMOABC with different methods of updating. In the employed bee stage, the normal food sources are updated as shown in Formula (11): (11)$$v_{ij}=x_{ij}+rand\left(0,1\right)\left(x_{nei,j}-x_{ij}\right)$$
where $v_{ij}$ represents the solution obtained after updating. $x_{nei,j}$ is the neighbor node with the largest degree among its neighbors in the scale-free network. The elite food sources are updated using
(12)$$v_{ij}=x_{ij}+rand\left(-1,1\right)\left(x_{ij}-x_{kj}\right)+rand\left(0,1\right)\left(x_{nei,j}-x_{ij}\right)$$
where $x_{kj}$ is a randomly selected SNP from the solution space, and k≠i. At the end of the employed bee stage, the onlooker bees select the food sources according to the fitness values. For the unselected solution, the onlooker bee repeats the operation of the employed bee stage to generate a new solution. The above process updates the food sources based on the scale-free network, and the search ability of the SFMOABC is improved effectively after updating. However, the ability to explore the unknown solution space needs to be further improved. Therefore, after getting an SNP combination $v_{i}$ by scale-free network updating, the opposition-based learning strategy is used to get another SNP combination $u_{i}$, which can be described as [35]
(13)$$u_{ij}=ub+lb-v_{ij}$$
where ub is the upper bound of the solution space, and lb is the lower bound of the solution space. The fitness values of $v_{i}$ and $u_{i}$ are calculated respectively, and the updated solution with a large fitness value is retained according to the greedy strategy. Then, the updated solution is compared with the initial solution $x_{i}$ through the greedy strategy again, and the solution with a large fitness value is retained as the final result of this iterative updating. The SFMOABC framework is shown in Figure 2.

#### 2.3.4. Time Complexity Analysis

The time complexity of the SFMOABC algorithm mainly depends on the construction of the scale-free network and the iteration of the ABC algorithm. The SFMOABC firstly needs to build a scale-free network in the initialization stage, and the total number of nodes in the network is consistent with the number of employed bees. The search process of the algorithm is guided by the scale-free network, and the structure of the scale-free network remains unchanged throughout the optimization process. In each iteration, the SFMOABC ranks the employed bees according to the quality of the food sources.

Here, we analyze the time complexity of the SFMOABC. T is the maximum number of iterations, and N is the number of employed bees. In the initialization stage, the time complexity of generating the initial population and calculating the fitness value is $O\left(2N\right)$, and the time complexity of constructing the network is $O(N^{2})$. Therefore, the time complexity of the initialization stage is $O(2N+N^{2})$. In the employed bee stage, the time cost of the SFMOABC sorting the employed bees according to their fitness values is $O(T\times N\log_{2}N)$, and the time complexity of the employed bees updating the food sources is O(N×T). Thus, the time complexity of the employed bee stage is $O((N+N\log_{2}N)\times T)$. In the onlooker bee stage, the SFMOABC first needs to calculate the probability of each food source being selected, and the time complexity is O(N×T). Then, the onlooker bees repeat the steps of the employed bee stage. Therefore, the time complexity of the onlooker bee stage is $O((2N+N\log_{2}N)\times T)$. The time complexity of the scout bee stage is O(N×T). According to the above analysis, the overall time complexity of the SFMOABC is $O((4N+2N\log_{2}N)\times T+2N+N^{2})$.

#### 2.3.5. Overall Framework

In order to solve the epistasis detection problem in GWAS, a multi-objective ABC algorithm based on the scale-free network is proposed. This includes three main parts: two objective functions, initialization based on the scale-free network, and solution updating based on the scale-free network. The SFMOABC combines the scale-free network and the opposition-based learning strategy into the ABC algorithm, which effectively increases the searching ability of the algorithm. At the same time, two complementary objective functions are used to make the results more accurate and reliable.

Algorithm 1 gives the pseudo-code of the SFMOABC. At the beginning of the algorithm, a population with N food sources is randomly generated, and each food source represents an SNP combination. Then, two objective functions are used to evaluate the quality of the SNP combinations, and the solutions are sorted in descending order according to their fitness values. After the initial population has been generated, the SFMOABC generates a scale-free network where the number of nodes is equal to the number of food sources. In the network, the $m_{0}$ hub nodes are numbered randomly, and the remaining $N-m_{0}$ nodes are numbered sequentially from $m_{0}+1$ according to the order that they joined the network. There is one-to-one correspondence between the solutions in the population and the nodes in the network. Since nodes in the network are divided into hub nodes and other nodes, the solutions in the population are also divided into two parts: elite food sources and normal food sources. Then, in the employed bee stage, the scale-free network and the opposition-based learning strategy are used to update the food sources, and the greedy strategy is used to select high-quality solutions. After the employed bee stage, the SFMOABC calculates the probabilities of the solutions based on their fitness values, where the bigger the fitness value, the greater the probability that onlooker bees will be chosen. When a food source is not selected, the onlooker bee will repeat the operation of the employed bee to generate a new solution and then keep a better one by the greedy strategy. In the scout bee stage, if a food source does not meet the condition of being replaced, the *trail* number is increased by 1. Otherwise, the *trail* number is reset to 0. Lastly, the model determines whether food sources should be abandoned by checking the *limit* parameter. If a food source cannot be improved further after a predetermined number of iterations, the food source will be abandoned, and a new food source will be generated randomly. Finally, SFMOABC iterates until the stopping condition is satisfied.

## 3. Experiments

### 3.1. Evaluation Measures

To avoid the one-sidedness of using a single evaluation indicator, two evaluation indicators, *Power* and *F-measure* [17], were used to evaluate the performance of the epistasis detection methods. Power is a measure of the ability to detect pathogenic models in all datasets; it can be expressed as
(14)$$Power=\frac{\#T}{\#S}$$
where #T is the number of datasets in which disease-related SNP combinations are successfully detected, and #S is the total number of simulated datasets generated with the same disease model (100 data matrices per disease model). The *F-measure* is a weighted average of the recall rate and accuracy rate, which can be defined as
**Algorithm 1:** SFMOABC**Input:** the number of the food sources N; the dimension of problems D; a count parameter representing the number of times the current solution has not been improved trail; the maximum number of not be improved limit; the number of hub nodes in the scale-free network m0; the number of edges when the node joins the network m.
**Output:** the optimal solution x. 01. Initialize N food sources to form a population X=x1,x2,⋯,xN;02. Calculate the fitness value of each solution F(x)=fx1,fx2,⋯,fxN;03. Build a scale-free network with N node, and number each node in the network.04. **While** *the stopping criteria is not satisfied*
**do**05.  **for** i=1→N
**do**06.   **if** i<=m0 **then**07.    vi=xi;08.    Find the neighbor nei with the largest degree of xi;09.    vi,d=xi,d+rand−1,1xi,d−xk,d+rand0,1xnei,d−xi,d10.    k∈1,2,⋯,N, k≠i; d∈1,2,⋯,D;11.    Calculate the fitness value of the vi.12.    vi,d′=ub+lb−vi,d;13.    Calculate the fitness value of vi′;14.    **if** fvi>fvi′ **then**15.     ui←vi; fui←fvi;16.    **else**17.     ui←vi′; fui←fvi′;18.    **end**19.    **if**
fui>fxi **then**20.     xi←ui; fxi←fui; traili←021.    **else**22.     traili←traili+1;23.    **end**24.   **else**25.    vi=xi;26.    Find the neighbor nei with the largest degree of xi;27.    vi,d=xi,d+rand0,1xnei,d−xi,d;28.    Repeat (Steps 10–23)29.   **end**30.  **end**31.  Calculate the probability Pi of xi;32.  i=1; t=0;33.  **while** t<N **do**34.   **if** rand>Pi **then**35.    repeat (Steps 06–30)36.   **end**37.   i←i+1;38.   **if** i←N+1 **then**39.    i←1;40.   **end**41.  **end**42.  Find the individual h with the maximum trail value;43.  **if** trialh>limit **then**44.   Randomly generated a new food source to replace the h-th food source;45.  **end****end**Return the optimal solution x with the largest fitness value.
(15)F−measure=21/recall+1/precision

High *recall* means that most of the truly associated SNP combinations are detected, but false positives may be detected as well. In contrast, high *precision* means that truly associated SNPs account for a large portion of the detected SNPs [36]. Recall can be written as
(16)$$recall=\frac{\#TP}{\#TP+\#FN}$$

Precision can be expressed as
(17)$$precision=\frac{\#TP}{\#TP+\#FP}$$

True positives (TPs) are defined as the discovery of a k-order SNP combination that is associated with disease status, false negatives (FNs) are defined as a nondiscovery of a SNP combination that is associated with disease, and false positives (FPs) are defined as a k-order SNP combination that is falsely associated with a disease status [17].

In this experiment, *#TP* is the number of datasets in which true disease-related SNP combinations were detected, *#FN* is the number of datasets in which no disease-related SNP combinations existed, and *#FP* is the number of datasets in which false disease-related SNP combinations were detected.

### 3.2. Simulation Data

Twelve commonly used two-order SNP interaction pathogenic models were selected to evaluate SNP combination detection identification methods, which were generated by the simulation software EpiSIM [37]. Models 1–8 are disease models with a marginal effect (DMEs), Models 9–12 are disease models without a marginal effect (DNMEs) [15,17]. For each model, 100 datasets were simulated, and each dataset contained 2000 cases and 2000 controls. In addition, there were 100 SNPs in small-scale datasets and 1000 SNPs in large-scale datasets. For each dataset, only one SNP combination was associated with the phenotype, whereas the others were not. The details of these models are shown in Table 1.

### 3.3. Parameter Settings

In order to prove its effectiveness, the SFMOABC was compared with AntEpiseeker [14], IACO [34], EpiACO [3], MACOED [38], SIPSO [16], BnBeeEpi [23], RCABC [24], and two single-objective ABC algorithms with MI and the K2-Score as objective functions (ABC_MI and ABC_K2). AntEpiseeker is a two-stage ACO algorithm. IACO is an improved ACO algorithm combining BN and MI. EpiACO is a single-objective ACO algorithm. MACOED is a multi-objective ACO algorithm combining logistic regression and BN. SIPSO is a particle swarm optimization (PSO) algorithm based on the scale-free network. BnBeeEpi is an improved ABC based on the Bayesian network. RCABC is a random-grouping-based self-regulating ABC algorithm.

In the simulation experiment, the number of iterations was set to 50 for all eight methods. The number of bees in the ABC algorithm, the number of ants in the ACO algorithm, and the number of particles in the PSO algorithm were set to the same value, which was 100 in the small-scale dataset and 1000 in the large-scale dataset. In the ABC algorithm, the parameter *limit* was set to 10. In BnBeeEpi, *honey.size* was set to 5 and *fast.α* was set to 0.001. In RCABC, *M* was set to 10 and the number of groupings was set to 15. In the ACO algorithm, the initial pheromone τ_0_ was set to 1, and the parameters α and β determining the weights of the pheromone and heuristic information were both set to 0.2. In the PSO algorithm, the acceleration factors c1 and c2 were both set to 2.05. In the scale-free network, the number of hub nodes $m_{0}$ was set to 3.

### 3.4. Experimental Results on Simulation Data

In the simulation experiment, the SFMOABC was compared with seven methods based on the swarm intelligence algorithm and two single-objective ABC algorithms (ABC_MI and ABC_K2) on 12 small-scale and 12 large-scale simulation models. The detection power values of the small-scale (100 SNPs) and large-scale (1000 SNPs) datasets are shown in Figure 3 and Figure 4. It can be seen from the results that the MACOED and AntEpiSeeker algorithms performed well with small-scale datasets, while their detection power values greatly reduced with large-scale datasets. In addition, the detection power of the SFMOABC based on the multi-objective method was much higher than that of the single-objective method under different models and scales. The experimental results also show that our method performed well and had good stability in different scale datasets, the situation that appeared in some other methods where the detection power decreased sharply with the expansion of the data scale was not observed.

The *F-measure* results of the eight methods for different models and scales are shown in Figure 5 and Figure 6. Moreover, we compared the running times of these seven methods based on the swarm intelligence algorithm and two single-objective ABC algorithms. We ran these methods independently 30 times on each epistatic model and took the logarithmic form of the average value of 30 independent experiments as the final result [39]. Figure 7 gives the running times of these eight methods.

Considering the detection power, F-measure, and running time comprehensively, SIPSO took less time, but its detection power was worse than the other methods. MACOED showed good detection power with small-scale datasets, but its performance was greatly reduced with large-scale datasets. This is because MACOED introduces the logistic regression to evaluate SNP combinations. This makes it unstable with different parameter settings, resulting in the search method being equivalent to a random search. The AntEpiSeeker method focuses on the identification of SNP interactions with marginal effects and, hence, shows the difference in the detection power among different sized datasets. The running times of MACOED and AntEpiseeker are much longer in different sized datasets. This is because the time complexity of these two methods is too high. BnBeeEpi showed good detection power in the different scale datasets, but since it utilizes the BN network structure to represent SNPs, it took a lot of time when the scale increased. RCABC required very little running time on the different scale datasets, because it applies distributed computing to the ABC algorithm, which greatly improves the running efficiency of the algorithm. The IACO method took less time and showed a higher accuracy level in small-scale datasets, but its detection power was low in large-scale datasets, especially in the model with no marginal effect. The SFMOABC comprehensively considers the utilization and development of the solution space. Therefore, compared with other methods, the SFMOABC has a shorter running time in different sized datasets, and it obtains high and stable detection power and F-Measure results. In other words, it can obtain more accurate results in less time, leading to better epistasis detection.

### 3.5. Experiment Results for Real AMD Data

To further verify the effectiveness of the SFMOABC, we conducted experiments on the AMD data [40]. We used AMD data including 50 control samples and 96 case samples, and each sample contained 103,611 genotypes of SNPs [32]. In the preprocessing stage, the K-nearest neighbor method was used to estimate the missing data [41]. There have been many studies on AMD disease, and the research results were used as a reference for our proposed algorithm. In this experiment, some detected SNP combinations that were highly correlated with AMD were output. According to fitness values, Table 2 lists the information related to the top 15 SNP combinations that may be associated with AMD obtained by the SFMOABC, including the names, genes, and chromosomes. The last two columns show the fitness values and *p*-values corresponding to the SNP combination. It can be seen that there is a negative correlation between the fitness value and the *p*-value.

The results show that most of the detected combinations contain rs380390 and rs1329428, two SNPs that are located on the *CFH* gene of chromosome 1. These SNPs have been widely reported to be associated with AMD [5,19,42,43,44,45,46]. SNP rs1363688 is located on the nongenetic coding region (N/A). It has also been shown to be associated with AMD [28,36,37]. SNP rs9328536 resides in the intron of the *MED27* gene. Transcription of the gene is triggered by factors that recognize transcriptional enhancer sites in DNA. In addition, *MED27* has been reported to be associated with melanoma [47], and thus mutations in the *MED27* gene may be associated with AMD. SNPs rs2402053, rs2380684, rs10512174, rs3913094, and rs724972 have been reported to be associated with AMD disease [43,48]. SNP rs10508731 resides in the *MPP7* gene on chromosome 10. The protein encoded by this gene plays a role in the establishment of epithelial cell polarity, and alternative splicing results in multiple transcript variants. It has been reported that *MPP7* can promote the migration and invasion of breast cancer cells through EGFR/AKT signaling [49]. SNP rs2466215 resides in the *PEBP4* gene of chromosome 8. *PEBP4* is a phosphatidylethanolamine (PE)-binding protein with key biological functions, and it has been reported that silencing of this gene may lead to complex diseases such as kidney disease and lung cancer [50]. These two SNPs are not mentioned in AMD-related literature, and hence, they need to be further studied to confirm whether they are truly associated with AMD.

Additionally, we used Cytoscape to build a visualized network of 2-order SNP combinations. The visualization results are shown in 8. The network contains 187 nodes and 182 edges. Each node in the network represents an SNP, and each edge connects two nodes, indicating that there is an association between the SNPs represented by these two nodes. As can be seen from 8, in addition to rs380390 and rs1329428, rs1740752 is also an important SNP, and many SNPs are associated with it. SNP rs1740752 is located in the noncoding region of human chromosome 10, as reported by Guo, et al. [43,44,48]. SNP rs2113379 is a variant in the *ADAM23* gene of chromosome 2. Members of this gene-encoded family are involved in various biological processes in cell–cell and cell–matrix interactions, including fertilization, muscle development, and neurogenesis [51]. It has been reported that the occurrence of common human cancers, such as gastric cancer, may be related to the inactivation of this gene [52]. In the Figure 8, the SNPs represented by the nodes marked in orange are associated with both rs380390 and rs1329428. Their associations may be the potential 3-order SNP combinations associated with AMD disease. Further work is needed to investigate the functions of these SNPs to test whether there are indeed 3-order epistatic interactions related to AMD disease.

## 4. Discussion

In this paper, the SFMOABC was proposed to detect epistasis in GWAS. We used two objective functions to make the results more accurate and more reliable. The key factor in the algorithm is the addition of the scale-free network to the solution updating process of the ABC algorithm. The scale-free network has the characteristics of a power-law distribution and low degree-degree correlation coefficient, and thus, the scale-free network can help the ABC algorithm to improve its search ability. In addition, the SFMOABC uses two objective functions to make the results more accurate and more reliable and introduces the opposition-based learning strategy to improve the randomness of the algorithm and maintain the diversity of the population. In this way, the exploration and exploitation abilities can be balanced, and the problems of converging too fast and falling into the local optimum can be avoided effectively. We tested the SFMOABC on 12 small-scale models and 12 large-scale models and compared it with SIPSO, IACO, epiACO, MACOED, AntEpiseeker, BnBeeEpi, RCABC, ABC_MI, and ABC_K2. The experimental results show that the SFMOABC is superior to the other methods. Finally, this method was applied to the real AMD data, and most of the SNP combinations found were proven to be associated with AMD disease in the corresponding literature. In general, the SFMOABC performs well in the epistasis detection of complex diseases. However, there are still some limitations. For example, only two-order epistasis can be detected at present. The problem needs to be further solved in future work.

## 5. Conclusions

### 5.1. Advantages

SFMOABC firstly adopts the scale-free network to guide the search of the ABC. It can help each employed bee to learn more effective information from its neighbors, which improves the detection power. Then, the multi-objective strategy in which the MI and K2-score are used to characterize various epistasis models and improve the detection power. what’s more, ABC can easily to fall into the local optimum as the iteration progresses, and the opposition-based learning strategy improves the detection power by improving the randomness of the algorithm.

### 5.2. Limitations

SFMOABC can only identify 2-order epistasis, but the occurrence of complex diseases is sometimes caused by the combined action of three or more SNPs. SFMOABC has not yet been able to identify these higher-order SNP combinations, and can only infer higher-order combinations from the results of 2-order combinations.

### 5.3. Future Work

In future research, we will focus on the detection of higher-order epistasis. The increase of the order will cause the algorithm to spend a lot of running time, so developing an effective method to detect high-order epistasis is something we need to consider in the future.

## Figures and Tables

**Figure 1 genes-13-00871-f001:**
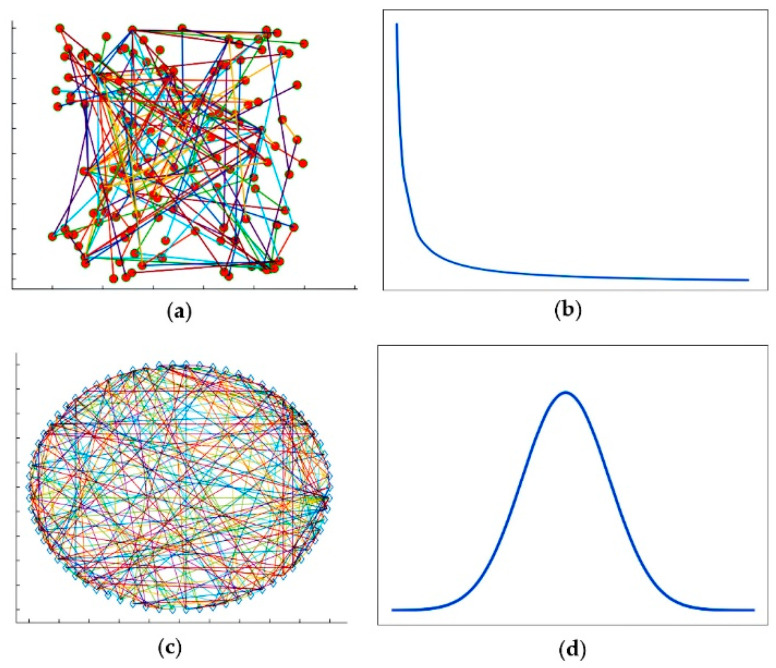
Comparison diagram of the scale-free network and the random network: (**a**) visualization of the scale-free network; (**b**) the degree distribution curve of nodes in the scale-free network; (**c**) visualization of the random network; (**d**) the degree distribution curve of nodes in the random network.

**Figure 2 genes-13-00871-f002:**
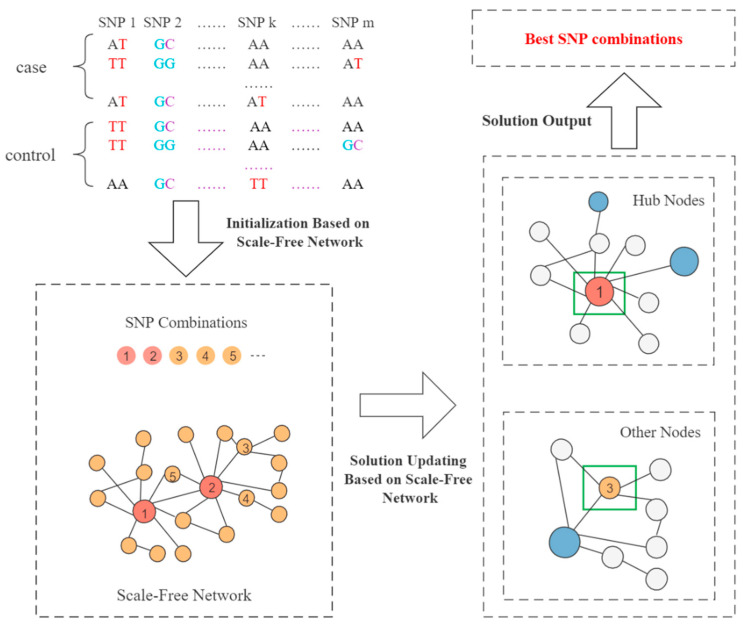
Framework of SFMOABC. The numbers in the figure represent the sequence numbers corresponding to the SNP combinations sorted in descending order according to the fitness value.

**Figure 3 genes-13-00871-f003:**
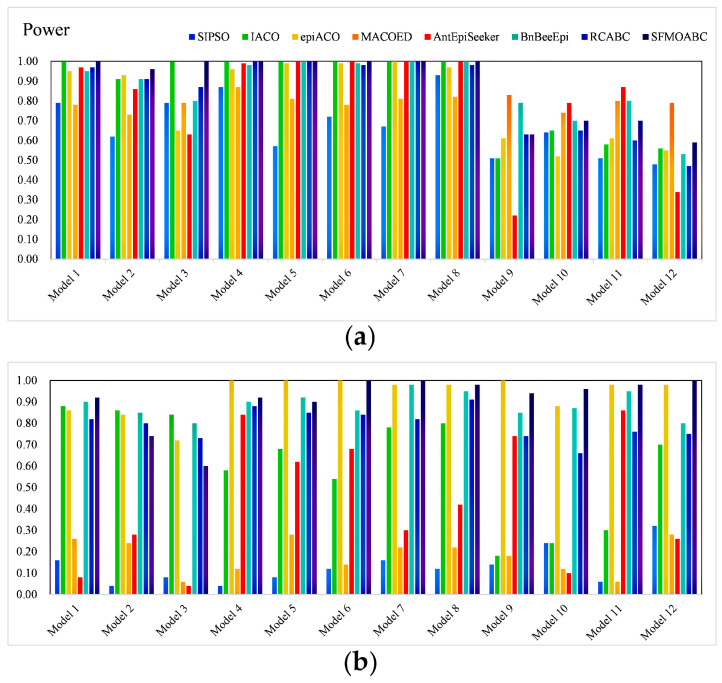
Power of methods based on the swarm intelligence algorithm: (**a**) Power on the small-scale datasets; (**b**) Power on the large-scale datasets.

**Figure 4 genes-13-00871-f004:**
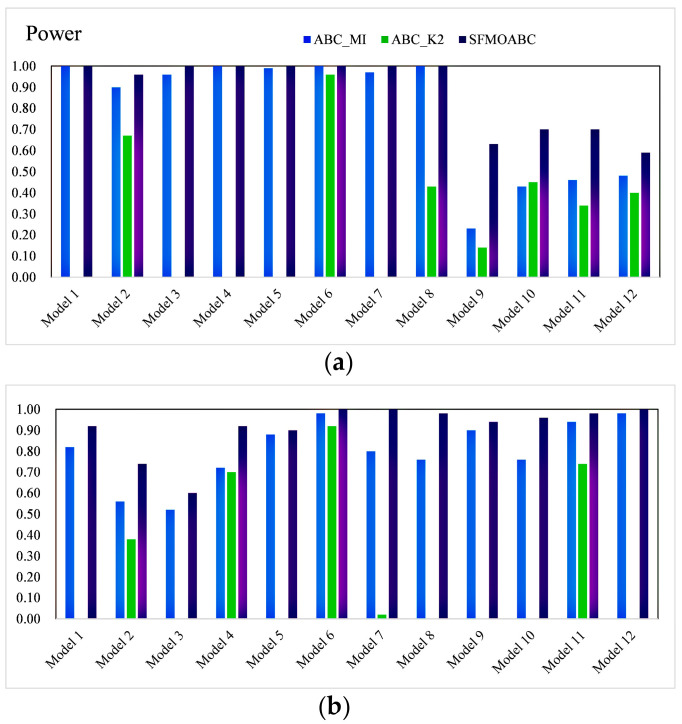
Power of the single-objective ABC algorithms: (**a**) Power on the small-scale datasets; (**b**) Power on the large-scale datasets.

**Figure 5 genes-13-00871-f005:**
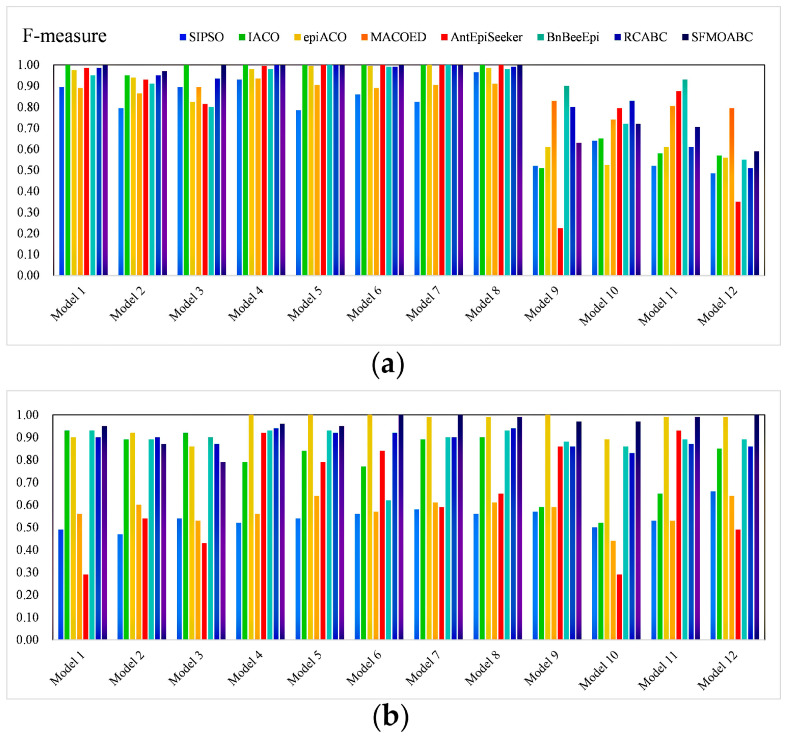
F-measure of methods based on the swarm intelligence algorithm: (**a**) F-measure on the small-scale datasets; (**b**) F-measure on the large-scale datasets.

**Figure 6 genes-13-00871-f006:**
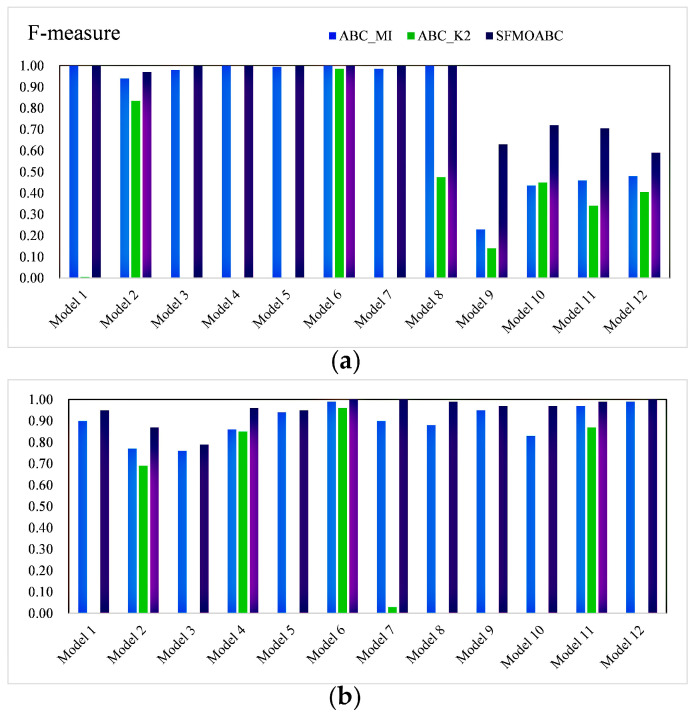
F-measure of single-objective ABC algorithms: (**a**) F-measure on the small-scale datasets; (**b**) F-measure on the large-scale datasets.

**Figure 7 genes-13-00871-f007:**
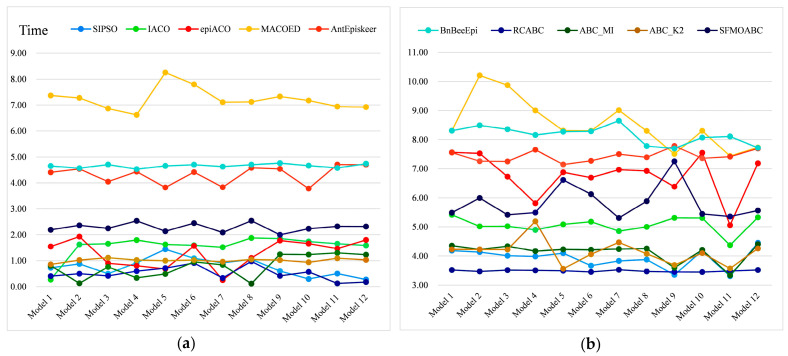
Running time: (**a**) Running time on the small-scale datasets; (**b**) Running time on the large-scale datasets.

**Figure 8 genes-13-00871-f008:**
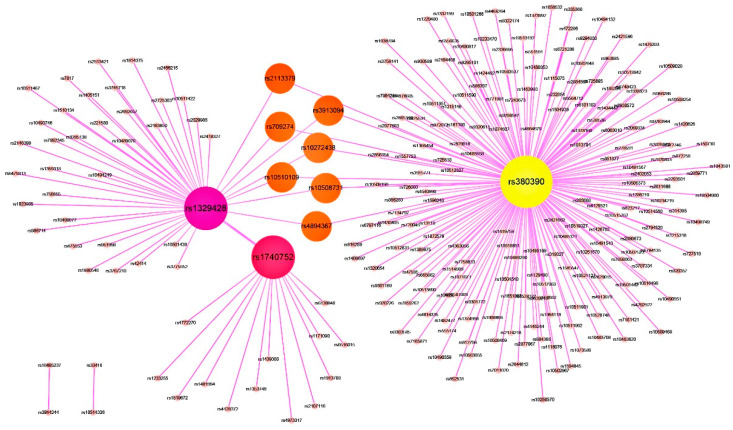
Epistasis network of AMD.

**Table 1 genes-13-00871-t001:** Details of the epistatic models.

Model	AABB	AABb	AAbb	AaBB	AaBb	Aabb	aaBB	aaBb	aabb
Model 1	0.087	0.087	0.087	0.087	0.146	0.190	0.087	0.190	0.247
Model 2	0.078	0.078	0.078	0.078	0.105	0.122	0.078	0.122	0.142
Model 3	0.009	0.009	0.009	0.013	0.006	0.006	0.013	0.006	0.006
Model 4	0.092	0.092	0.092	0.092	0.319	0.319	0.092	0.319	0.319
Model 5	0.084	0.084	0.084	0.084	0.210	0.210	0.084	0.210	0.210
Model 6	0.052	0.052	0.052	0.052	0.137	0.137	0.052	0.137	0.137
Model 7	0.072	0.164	0.164	0.164	0.072	0.072	0.164	0.072	0.072
Model 8	0.067	0.155	0.155	0.155	0.067	0.067	0.155	0.067	0.067
Model 9	0.486	0.960	0.538	0.947	0.004	0.811	0.640	0.606	0.909
Model 10	0.103	0.063	0.124	0.098	0.086	0.069	0.021	0.147	0.059
Model 11	0.000	0.000	0.000	0.000	0.050	0.000	0.100	0.000	0.000
Model 12	0.000	0.020	0.000	0.020	0.000	0.020	0.000	0.020	0.000

**Table 2 genes-13-00871-t002:** Top 15 Captured Epistatic Interactions Associated with AMD.

SNP1			SNP2			Fitness Value	*p*-Value
Name	Gene	Chr	Name	Gene	Chr		
rs380390	*CFH*	1	rs1363688	*N/A*	5	39.16	1.5453 × 10^−9^
rs380390	*CFH*	1	rs2402053	*N/A*	7	37.72	1.4679 × 10^−8^
rs380390	*CFH*	1	rs1374431	*LOC107985962*	2	37.19	2.6240 × 10^−8^
rs1329428	*CFH*	1	rs9328536	*MED27*	9	36.81	3.0901 × 10^−8^
rs380390	*CFH*	1	rs2380684	*N/A*	2	36.00	3.9086 × 10^−8^
rs380390	*CFH*	1	rs3009336	*N/A*	1	34.59	5.3535 × 10^−8^
rs380390	*CFH*	1	rs555174	*N/A*	21	34.14	5.7995 × 10^−8^
rs380390	*CFH*	1	rs2794520	*N/A*	1	33.56	6.7417 × 10^−8^
rs380390	*CFH*	1	rs10508731	*MPP7*	10	33.56	7.2188 × 10^−8^
rs380390	*CFH*	1	rs1740752	*N/A*	10	33.50	1.0917 × 10^−7^
rs1329428	*CFH*	1	rs10489076	*N/A*	4	33.43	1.9228 × 10^−7^
rs1329428	*CFH*	1	rs3913094	*N/A*	12	33.16	4.2460 × 10^−7^
rs380390	*CFH*	1	rs724972	*N/A*	3	33.02	4.4484 × 10^−7^
rs1329428	*CFH*	1	rs724972	*N/A*	3	33.02	1.3223 × 10^−6^
rs1329428	*CFH*	1	rs2466215	*PEBP4*	8	32.35	1.7865 × 10^−6^

## Data Availability

Not applicable.

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
