# Peer review of "Multi-Objective Artificial Bee Colony Algorithm Based on Scale-Free Network for Epistasis Detection"

_genes, 2022, doi:10.3390/genes13050871_

Round 1
Reviewer 1 Report
The authors presented a method for epistasis detection using a multi-objective artificial bee colony algorithm. The manuscript is well written and interesting. However, there are a few shortcomings that should be incorporated:
- The authors should endeavour to define acronyms in the first instance. For example, SNP is not defined before it is used in the abstract.
- The second paragraph in the introduction should be split into multiple paragraphs.
- Most of the citations in the manuscript are not properly done, for example, Wan et al., Wang et al., does not seem correct.
- In equation 4, rand(-1,1) is not defined.
- In lines 160-161, the authors stated that "H (S ) is the entropy of S,
H (S ) is the entropy of Y ", kindly correct this. - Equation 7 does not seem to be correct, if H(S;Y) represents the joint entropy of S and Y , then the addition (+) in the equation should be a minus (-). Kindly refer to the following: https://en.wikipedia.org/wiki/Mutual_information https://link.springer.com/chapter/10.1007/978-3-030-96308-8_49
- In lines 297-287, the explanations for recall do not seem clear, the same thing with True positive, True Negative, False positive, and False Negative. Kindly define them properly.
- The authors can make the discussion more robust by comparing their approach with some state-of-the-art methods in recent literature.
Reviewer 2 Report
The authors have proposed a multi-objective ABC algorithm based on the scale-free network for epistasis detection. It is a good paper, and well structured. However, the following comments are made:
1) The introduction does not stimulate to go ahead with the remaining of the paper because it does not introduce really new topic/solution, since the artificial bee colony algorithm has been utilized before for Epistasis detection.
Please see:
BnBeeEpi: An Approach of Epistasis Mining Based on Artificial Bee Colony Algorithm Optimizing Bayesian Network-2019 (10.1109/BIBM47256.2019.8983151)
A random grouping-based self-regulating artificial bee colony algorithm for interactive feature detection-2022 (10.1016/j.knosys.2022.108434)
We recommended the authors at least compare their proposed method with the above-mentioned ABC-based techniques.
2) Furthermore, "the research motivation and challenges" and "the main contribution…" of the paper in the introduction section are missing. Please rewrite this section.
3) Besides, please provide a section named "Related works" to review some of the recently proposed works (most of the mentioned works are old) about epistasis detection using meta-heuristic-based techniques, while some of the papers that should have been included are:
https://doi.org/10.3390/genes12020191
10.1109/TCBB.2020.3030312
10.1109/ACCESS.2019.2894676
https://doi.org/10.1016/j.knosys.2022.108434
https://doi.org/10.1016/j.ab.2021.114242
4) What is the overhead (time complexity) proposed solution? Please provide a subsection to discuss the overhead Algorithm.
5) In the experimental section, statistical tests of the hypothesis should be used to determine whether the differences shown in the figures and obtained results are statistically significant or due to chance.
6) We also suggested the authors add a final generated network diagram for the real dataset.
Round 2
Reviewer 2 Report
The manuscript has been significantly improved. The authors have responded to my questions quite well. Thanks.